# Genetic Variants of Lipoprotein Lipase and Regulatory Factors Associated with Alzheimer’s Disease Risk

**DOI:** 10.3390/ijms21218338

**Published:** 2020-11-06

**Authors:** Kimberley D. Bruce, Maoping Tang, Philip Reigan, Robert H. Eckel

**Affiliations:** 1Division of Endocrinology, Metabolism, and Diabetes, Department of Medicine, University of Colorado Anschutz Medical Campus, Aurora, CO 80045, USA; maoping.tang@cuanschutz.edu (M.T.); robert.eckel@cuanschutz.edu (R.H.E.); 2Department of Pharmaceutical Sciences, Skaggs School of Pharmacy and Pharmaceutical Sciences, University of Colorado Anschutz Medical Campus, Aurora, CO 80045, USA; philip.reigan@cuanschutz.edu

**Keywords:** lipoprotein lipase, Alzheimer’s disease, microglia, lipoproteins, apoproteins

## Abstract

Lipoprotein lipase (LPL) is a key enzyme in lipid and lipoprotein metabolism. The canonical role of LPL involves the hydrolysis of triglyceride-rich lipoproteins for the provision of FFAs to metabolic tissues. However, LPL may also contribute to lipoprotein uptake by acting as a molecular bridge between lipoproteins and cell surface receptors. Recent studies have shown that LPL is abundantly expressed in the brain and predominantly expressed in the macrophages and microglia of the human and murine brain. Moreover, recent findings suggest that LPL plays a direct role in microglial function, metabolism, and phagocytosis of extracellular factors such as amyloid- beta (Aβ). Although the precise function of LPL in the brain remains to be determined, several studies have implicated LPL variants in Alzheimer’s disease (AD) risk. For example, while mutations shown to have a deleterious effect on LPL function and expression (e.g., N291S, *HindIII*, and *PvuII)* have been associated with increased AD risk, a mutation associated with increased bridging function (S447X) may be protective against AD. Recent studies have also shown that genetic variants in endogenous LPL activators (ApoC-II) and inhibitors (ApoC-III) can increase and decrease AD risk, respectively, consistent with the notion that LPL may play a protective role in AD pathogenesis. Here, we review recent advances in our understanding of LPL structure and function, which largely point to a protective role of functional LPL in AD neuropathogenesis.

## 1. Introduction

Alzheimer’s disease (AD) is a devastating and ultimately fatal age-associated neurodegenerative disease. Currently, 5.8 million Americans are living with AD. Of these individuals, 81% are 75 or older [1]. In fact, it has been estimated that 32% of individuals that reach the age of 85 will have AD [1]. AD is characterized by increased deposition (or impaired clearance) of amyloid-beta (Aβ) deposits, the formation of neurofibrillary tangles, tau protein, and neuroinflammation [2]. However, all drugs currently approved by the US Food and Drug Administration (FDA) for the treatment of AD modulate neurotransmitters, and do not target the evolution of the disease. The current standard of care for AD includes cholinesterase inhibitors (ChEIs, e.g., donepezil) and partial *N*-methyl-D-aspartate (NMDA) antagonists (e.g., memantine), or a combination therapy (e.g., Namzaric). While these treatments may address some of the primary and secondary symptoms of AD, they are unable to prevent or delay AD onset or progression. Therefore, there is an urgent medical need for the design of disease-modifying therapeutics (DMTs) that specifically target the underlying mechanisms of AD neuropathogenesis to improve the quality of life and clinical outcomes for patients with, and at risk for, AD. To develop such therapeutics, novel targets with the potential to modify AD risk require identification.

Lipoprotein lipase (LPL) is a secretory protein primarily involved in the hydrolysis of triglycerides (TG) in TG-rich chylomicrons and very-low-density lipoproteins (VLDL). Therefore, LPL has traditionally been a mechanistic target for the development of TG lowering medications to ameliorate cardiovascular disease (CVD). However, previous work from our laboratory has shown that LPL is expressed in several regions of the central nervous system (CNS), including the brain, spinal cord, and peripheral nerves [3,4,5]. In addition, recent RNA-seq studies have shown that LPL is predominantly expressed by macrophages and microglia in the human brain, and expressed by macrophages, microglia, and oligodendrocyte precursor cells (OPCs) in the murine brain [6,7]. Importantly, LPL has been repeatedly implicated in AD pathogenesis. Specifically, loss-of-function LPL polymorphisms have been linked to increased AD risk [8]. Whereas patients with a gain-of-function LPL polymorphism (S447X) have increased LPL activity, lower VLDL-TG, and reduced amyloid plaque formation [9]. There is a growing body of literature highlighting the importance of LPL in microglial function [10,11,12,13]. Indeed, research from our laboratory has recently shown that LPL is a feature of anti-inflammatory microglia, which are involved in Aβ phagocytosis and lipid uptake [12,13]. Although studies aimed at elucidating the function of LPL in the CNS are ongoing, since lipid and lipoprotein metabolism in the CNS plays a major role in AD risk, it is plausible that LPL contributes to AD pathology via these processes. In this review, we detail recent advances in our understanding of LPL function and biology, and critically evaluate evidence supporting a genetic association between LPL and AD. In this article, we are the first to map LPL variants to a recent crystal structure of LPL [14], and review recent studies investigating the functional significance of these variants. Moreover, we discuss emerging evidence suggesting that factors that regulate LPL function and processing may also contribute to AD risk, further highlighting the role of LPL in the neuropathogenesis of AD.

## 2. LPL: Structure, Function, and Regulation

The human LPL gene (synonyms include LIPD and HDLCQ11) is found on chromosome 8p21.3, and contains 10 exons and 11 introns, and encodes a 475 amino acid protein (Figure 1). It is important to note that the base numbering has been updated since initial characterization and modeling of the LPL enzyme, resulting in a discrepancy between the numbering of the amino acids modified in a given variant. For example, the LPL variant Asn291Ser, (rs268) is actually found at position 318, and is therefore also known as Asn318 (Figure 1). Similarly, variant S447X is actually at position 474 (Figure 1). These inconsistencies have arisen because of a change in the number of the bases in the LPL gene during human reference genome Build 37, and, due to updated models for LPL structure (see below). There is also a discrepancy in the literature regarding the number of exons contained within the protein, which may reflect splice variants, however, the functional significance of these has not yet been determined. Interestingly, many mutations in the LPL gene are found in exons 4,5, and 6, including missense, nonsense, frameshift, insertion, and duplication mutations [15].

The LPL enzyme can be catalytically active as a homodimer composed of two glycosylated 55 kDa subunits connected by non-covalent interactions in a ‘head-to-tail’-orientation [16,17]. Structural homology modeling has indicated that LPL has a similar structure to pancreatic lipase (PL) [18]. However, recent crystallography studies have improved our understanding of LPL structure and function. The crystal structure of LPL (in complex with glycosylphosphatidylinositol-anchored high-density lipoprotein binding protein 1 [GPIHBP1]) was recently resolved, and has shown that LPL can be active as a monomeric 1:1 (LPL:GPIHBP1) complex [19]. In addition, the crystal structure of LPL:GPIHBP1 with an inhibitor bound to the active site of LPL, has provided the first crystal structure where the lid and lipid-binding regions necessary for TG-rich lipoprotein (TRL) recognition are visible [14]. Here, we have annotated the structure from Arora et al., with mutations that alter LPL function (Figure 1).

Although the regulation of LPL activity is not fully understood in the CNS, in the periphery, catabolism of TRL particles is known to be dependent on a number of factors that regulate LPL processing (Figure 2). For example, Lipase Maturation Factor 1 (LMF1) is a chaperone protein of the endoplasmic reticulum that is required for folding and secretion of LPL [20]. In addition, LMF1 facilitates the formation of the correct intermolecular disulfide bonds in the LPL protein [21]. Following secretion (from heart, muscle, and adipose tissue cells), LPL is tethered to heparan sulfate proteoglycans (HSPGs) in the endothelium [14], where it acts to liberate (FFAs) for uptake by key metabolic tissues. However, LPL can easily dissociate from these sites and re-associate with cell surface receptors and act as a molecular bridge to facilitate uptake and catabolism of lipoproteins [22]. These include receptors involved in lipoprotein endocytosis, such as the LDL receptor-related protein/α2-macroglobulin receptor (LRP), GP330/LRP-2 (Megalin), and VLDL receptors [23]. In addition, LPL binds to Sortilin with a similar affinity as LRP. Moreover, this interaction can be inhibited by the receptor-associated protein (RAP) and neurotensin, suggesting that LPL and RAP compete for interaction with Sortilin [24]. Since RAP-deficient adipocytes secrete poorly assembled LPL, it has been suggested that RAP may prevent premature interaction of LPL with binding partners in the secretory pathways such as LRP and HSPGs [25] (Figure 2).

In addition to factors associated with trafficking and processing, there are several endogenous proteins associated with LPL inhibition. The angiopoietin-like proteins (ANGPTL) 3,4, and 8 inactivate LPL activity by catalyzing the unfolding of LPL’s hydrolase domain [26,27] (Figure 2). ANGPTL4 is one of the best characterized LPL inhibitors and has recently been shown to unfold the catalytic domain of LPL in sub-stoichiometric amounts [28]. Importantly, the binding of GPIHBP1 to LPL protects LPL from ANGPTL4-mediated unfolding [28]. ANGPTL3 inhibits LPL activity in vitro and can also inhibit endothelial lipase, which catalyzes the hydrolysis of phospholipids in circulating lipoproteins, mostly high-density lipoproteins (HDL) [29,30]. Importantly ANGPTL3-mediated inhibition of LPL in vivo requires activation by ANGPTL8, which is similarly located in the intron of a DOCK (dedicator of cytokinesis) gene [31], suggesting divergent functions emerging through evolution.

The apolipoproteins (Apo) C-I and C-III are also known to inhibit LPL activity (Figure 2). However, unlike the ANGPTLs, ApoC-I, and ApoC-III are thought to prevent LPL from binding to lipid particles, thus negatively regulating LPL-activity and rendering the enzyme more susceptible to irreversible inactivation by ANGPTL4 [32] (Figure 2). APOE is another apoprotein associated with negative regulation of LPL. Specifically, APOE inhibits LPL-mediated lipolysis of chylomicron-like emulsions in vivo and in vitro [33]. Importantly, APOE is expressed at extremely high levels in the brain, and some isoforms of APOE (APOE4) are associated with the neuropathogenesis of AD. However, a role for APOE in the regulation of LPL in the CNS has not been investigated.

In addition to endogenous inhibitors, there are also endogenous activators of LPL activity. ApoA5 increases LPL activity and has been proposed to direct VLDL and chylomicrons to proteoglycan bound-LPL [34]. ApoC-II is a promiscuous apoprotein constituent of all lipoproteins, including very low-density lipoproteins (VLDL) that are needed for LPL activation [35]. Interestingly, a synthetic fragment of ApoC-II (residues 55-78) is capable of activating LPL activity 12-fold, compared to 13-fold for the intact protein [36]. Also, of note is the fact that ApoC-II is predominantly expressed in the macrophage and microglia of both the human and mouse brain [6]. However, in the human brain, ApoC-II is also expressed by oligodendrocytes and mature astrocytes [6,7]. A putative role for ApoC-II in the brain has not been investigated.

Other proteins within the apoproteins family play a role in lipid and cholesterol homeostasis in the brain, and therefore it is plausible to speculate that they may directly or indirectly interact with LPL. For example, clusterin (CLU), which is also known at ApoJ, is considered the third greatest risk factor for late-onset AD (LOAD), following ApoE and BIN1 [37,38]. CLU’s role in Aβ uptake and clearance most likely underlies its association with AD neuropathogenesis. Although there have been reports that CLU facilitates [39] and impedes Aβ clearance [40], it is argued that the outcome of the interaction relates to the CLU: Aβ ratio, and that either factor in excess determines a neuroprotective or neurotoxic outcome [41,42]. Recently, it has been suggested that CLU and ApoE are both involved in the lipidation and triggering receptor expressed on myeloid cells (TREM2)-mediated clearance of Aβ by microglia [43]. Since LPL and TREM2 are co-expressed in microglia [11], and both factors are involved in lipoprotein processing, it is likely that CLU and TREM2 functionally contribute to AD pathology. However, a direct interaction has yet to be reported and further studies are warranted.

## 3. LPL in the Central Nervous System

LPL is expressed in the brain, spinal cord, and peripheral nerves [3,4,5], and is predominantly expressed by macrophages and microglia in the human and murine brain [6]. We have previously shown that neuronal-LPL is associated with the central regulation of systemic metabolism [44,45]. Mice lacking neuronal LPL become obese and have a specific depletion in polyunsaturated fatty acids (PUFAs) [45,46]. In addition, hypothalamic neurons lacking LPL show altered substrate metabolism in vitro and in vivo, and systemic substrate metabolism in vivo [47]. In primary neurons, LPL activity has been shown to be negatively regulated by the sortilin-related receptor (SorLA/sortilin) [48]. Specifically, SORL1, which encodes SorLA (further processed to sortilin), expression is associated with increased accumulation of LPL within endosomal compartments, which is ultimately routed to the lysosomes for degradation and prevents LPL from being secreted [48].

Several studies have also demonstrated putative roles for LPL in glia. For example, LPL expression is increased in primary cultures of hypothalamus-derived astrocytes, whereas palmitic acid and TGs reduce LPL expression ex vivo [49]. Moreover, astrocyte-specific depletion of LPL in vivo, exacerbates high-fat diet-induced obesity, suggesting that astrocyte-derived LPL is involved in central homeostasis and peripheral metabolism [49]. LPL can bind to Aβ, which promotes glycosaminoglycan-dependent uptake of Aβ by astrocytes [50]. Interestingly, ANGPTL4 is abundantly expressed in white matter astrocytes, and its expression is markedly reduced in active multiple sclerosis (MS) lesions [51].

Recent data suggests that LPL regulates microglial metabolism and phagocytosis, which may underlie a role in the development of AD. Several single-cell RNAseq (scRNAseq) analyses of microglia ex vivo, have defined the transcriptomic identities of microglial phenotypes throughout aging, across brain regions [52,53,54], and during disease [11,53,55]. A microglial phenotype with tightly regulated lipid and lipoprotein metabolism has been consistently observed. For example, Keren-Shaul et al., 2017 observed disease-associated microglia (DAMs) in aged 5XFAD mice characterized by markedly increased expression of genes associated with lipid and lipoprotein metabolism (e.g., APOE, LPL, TREM2), suggesting that lipid uptake, utilization, or metabolism is increased in these microglial sub-populations [11]. Interestingly, elevated LPL expression is also a feature of microglial clusters observed during early postnatal life (P4/P5) [53], in proliferative-region-associated-microglia (PAMs) [54], in the demyelinating brain [53], and in the latter stages of neurodegeneration [55]. While its precise function remains elusive, we have previously shown LPL is a feature of phagocytic and ‘reparative’ microglia and is involved in microglial lipid and lipoprotein uptake [1]. LPL depletion in BV-2 microglia (LPL KD) also causes a profound inflammatory polarization [1]. Taken together, these data suggest that LPL is involved in lipid and lipoprotein uptake, which may provide the lipid substrates needed for homeostatic microglial functions such as oxidative metabolism and phagocytosis.

## 4. LPL Variants and AD

Polymorphisms in the LPL gene that encode non-functioning LPL proteins lead to LPL deficiency, also known as familial chylomicronemia syndrome (FCS) [56]. FCS is a rare (1 in 1 million) autosomal recessive disorder and involves impaired clearance of chylomicrons and VLDL from plasma leading to serious clinical consequences such as acute and recurring pancreatitis which can be fatal [56]. Although clinical investigations regarding the association between familial LPL deficiency and AD are lacking, neuropsychiatric findings have been reported, such as depression, memory loss, schizophrenia [57], and dementia.

The N291S (rs268) polymorphism occurs in exon 6 of the LPL gene and results in an A-G substitution (Table 1). In silico analysis suggests that this substitution leads to changes in the secondary structure of LPL, and changes in the hydrophobicity and flexibility of residues 314-322 near the ligand-binding site, which is detrimental to LPL function [58]. Several studies have reported that individuals with the N291S allele have higher circulating TGs but lower HDL-cholesterol (HDL-C) [59,60,61,62]. While the prevalence of N291S is around 1% of the general population, N291S is over-represented (5%) in patients diagnosed with AD, suggesting that the loss of function LPL variant plays a role in AD pathogenesis [63]. Computational analysis suggests that the amino acid substitution that occurs at position 318 in the N291S variant results in changes the hydrophobicity and flexibility of residues 314–322 near the ligand binding site [58]. Although studies investigating prevalence in relatively small cohorts have reported no association with N291S and AD risk, larger analysis and meta-analysis have reported that N291S is a predisposing factor of AD [8].

The *HindIII* (rs320) variant is located in intron 8, and removes the *HindIII* restriction enzyme recognition site (Table 1) [70]. The presence of the *HindIII* variant has been shown to impair transcription factor binding, leading to decreased LPL expression in vitro [71]. Specifically, the *HindIII* site is a putative TATA box, and subunits of TFIID and TATA-binding protein (TBP) have a reduced affinity for the variant sequence [71]. Moreover, *HindIII*(+/+) individuals have an increased risk of stroke, coronary artery disease, elevated levels of plasma TG, and reduced HDL-C [72]. However, recent analysis has suggested that *HindIII* (+/−) carriers may be protected against ischemic stroke (IS) [73]. In an Italian population, *HindIII* (+/+), but not *HindIII*(+/−) was associated with an increased risk of AD [66]. In an Iranian cohort, the frequency of the *HindIII* (+/+) genotype was significantly higher in patients with LOAD than controls [65]. Indeed, there was a 1.75-fold increase of LOAD in *HindIII* (+/+), which was more pronounced in males [65].

The *PvuII* (rs285) SNP is found in intron 6 of the LPL gene and is associated with increased TG, HDL-C, diabetes, and coronary artery disease incidence and severity (Table 1) [67,74]. Screening of a large eastern Canadian cohort of autopsy-confirmed control and AD brains demonstrated a significant association between *PvuII* and sporadic AD [67]. Importantly, *PvuII* was associated with reduced cortical cholesterol concentrations, increased neurofibrillary tangles (NFTs), and increased senile plaques [67]. In addition, there was an interaction between senile plaque formation and the APOE4 allele in the fusiform gyrus [67]. While the effect of an intrinsic variation on protein function is unlikely, the fact that the *PvuII* restriction site is a non-coding cis-regulatory element is likely of functional significance. However, in silico analysis of *PvuII* suggest that this variant does not affect the splicing pattern of the LPL hnRNA, or hnRNA structure or stability [58].

Variant Ser447Ter (rs328), also known as S447X, occurs in exon 9 of the LPL gene and leads to a C-G transversion and a truncated version of the LPL protein that lacks the two final amino acids (Table 1). It has been suggested that the truncation of these amino acids in the *C*-terminal leads to exposure of amino acids 401–413, which represent putative lipoprotein receptor binding sites, and represent a mechanism of action of increased lipoprotein uptake and reduced CAD risk in individuals carrying this variant [68]. Unlike the mutations already discussed in this review, the S447X variant is associated with reduced TG, and reduced risk of developing CAD and T2D; indicative of a gain-of-function mutation [75,76]. However, the mechanism underlying this increased function is not fully understood. On one hand, in vitro studies using COS-1 cells to express the S447X variant have shown increased hydrolytic activity compared to WT (185%) when ^3^H triolein was used as a substrate [77]. In contrast, in vitro studies have also demonstrated no differences in the hydrolytic activity of LPL harboring the S447X variant (LPL^S447X^), or any greater susceptibility to ANGPTL4 mediated inhibition [68]. However, LPL^S447X^ is associated with increased LPL-mediated uptake of fluorescently labeled lipoprotein particles, suggesting that the truncation exposes residues involved in receptor binding and endocytosis [68]. In addition, these studies have reported both an increase in LPL mass (131%) [78], and an increase in protein translation, due to reduced translation inhibition associated with the S447X variant [79]. Therefore, an increased abundance of LPL in individuals carrying the S447X mutation may account for the clinical benefit of this variant. However, it is also likely that major differences in the methods used to measure LPL activity may account for discrepancies in the literature regarding the functional effects of the S447X variant.

Due to its therapeutic potential, the S447X variant has been successfully administered in mice, and was the first approved gene therapy in humans (alipogene tiparvovec/glybera) aimed at reducing TG and alleviating symptoms (i.e., recurrent pancreatitis) in patients with LPL-deficiency. However, this drug was later pulled from the market due to the high cost of manufacturing. Interestingly, the S447X mutation is one of the most common and is found in 5–12% of the general population, with variances depending on the ethnic group [63]. Although there have been reports of a lack of association between the S447X mutation and AD risk [80], additional studies, including two meta-analyses indicating that LPL S447X may be underrepresented in patients with AD and therefore is a protective factor of AD [8,81]. Since the S447X is associated with increased lipoprotein uptake, it is, therefore, plausible to speculate that increased clearance of lipoproteins in the brain may contribute to the possible protection from AD. In addition, increased LPL^S477X^-mediated endocytosis of APOE4 containing astrocyte-derived HDL may be protective in aging and disease. However, since CAD and AD share many clinical characteristics [82], the contribution of the S447X mutation to reduced AD risk may also be a result of indirect effects on peripheral metabolism.

## 5. Genetic Variants Regulating LPL Function and Processing and AD Risk

Altered LPL-mediated lipid and lipoprotein processing in the CNS, and periphery may contribute to AD risk. Therefore, it is not surprising that genetic (and epigenetic) variants of the proteins involved in LPL regulation and processing (discussed in detail above) are also associated with changes in AD risk.

The best-characterized genetic determinant of familial AD (FAD) and LOAD is the apolipoprotein E (ApoE) allele epsilon 4 (APOE4). It is thought that this variant plays several roles in the neuropathogenesis of AD, including alterations in microglial lipid and lipoprotein metabolism and microglial function leading to reduced clearance of Aβ [83]. The role of APOE4 in the pathobiology of AD has been extensively reviewed elsewhere [84]. However, although an interaction between APOE4 and the potential inhibition by APOE-containing lipoproteins on LPL-mediated lipoprotein metabolism in the brain is plausible, this has not yet been directly studied.

There are only a few reports of genetic variants in the ApoC family and AD risk. The Taq1 “F allele” of the ApoC-II gene is positively associated with Alzheimer’s dementia [85]. Further analysis has revealed that the ApoC-II F allele is positively associated with late-onset AD (LOAD), but not early-onset FAD [86]. In addition, an interaction between ApoC-II and APOE in late-onset FAD has been described [87]. Since ApoC-II is a known LPL activator, the association between a dysfunctional ApoC-II variant and AD suggests reduced LPL-mediated lipid processing; however, this remains to be studied. Consistent with this hypothesis, it has been noted that the rare ApoC-III “G allele” variant may offer some protection against the development of sporadic AD (SAD) in a Chinese population [88].

Several recent studies have highlighted a role for SORL1 in the development of both late-onset and early-onset AD [89,90]. Specifically, SORL1 expression is reduced in SAD [91]. Since SORL1 is involved in neuronal endosomal trafficking, it is thought that functional variants in SORL1 lead to defects in Aβ and APP processing [92,93]. Moreover, loss of function mutations are thought to cause LOAD [94]. Since SORL1 is also associated with LPL trafficking and secretion, it would be of interest to determine whether there was a synergistic interaction between these two proteins regarding AD risk. However, SORL1 is involved in the intracellular trafficking of many factors, including enzymes, growth factors, and signaling receptors, and is not specific to LPL. Therefore, LPL-specific interactions may be challenging to discriminate [95].

## 6. LPL Variants and Other CNS Disorders

LPL polymorphisms have also been implicated in other CNS disorders such as vascular dementia and ischemic stroke (IS) [58,73]. While the mechanisms may be distinct from the development of AD, evaluating the role of gain, and loss-of-function mutations may shed light on the etiology of the disease and the function of LPL in the CNS. For example, recent studies suggest that individuals with the *HindIII(+/+)* and *PvuIII* mutations, which are thought to be detrimental to LPL function, may be protected against IS [65,73]. In contrast, the “gain-of-function” Ser447Ter polymorphism is not associated with a reduced risk of IS, suggesting that LPL-mediated lipoprotein uptake may exacerbate stroke risk [65]. However, inconsistent observations have been reported [72], prompting the need for further study.

LPL-dependent regulation of lipid and lipoprotein metabolism in the brain may influence neuronal function and metabolism, astrocyte metabolism, microglial lipid metabolism, phagocytosis, and immunomodulation (gliosis). Therefore, altered expression, processing or function of LPL has the potential to underlie the pathogenesis of many CNS disorders such as AD, MS, Parkinson’s disease, and beyond. While associations with LPL polymorphisms and AD have been reported, there have been no reports (at the time of publication of this manuscript) that link LPL mutations with any other neurodegenerative disease. However, since altered abundance of LPL has been repeatedly implicated in AD and demyelinating disorders, metabolic diseases, and aging, it is likely that further study may reveal additional associations with functional LPL mutations.

## 7. Conclusions

In summary, there have been a number of reports linking LPL polymorphisms to increased (loss of function mutations [N291S]) or reduced (increased LPL function [S447X]) AD risk. While the mechanism of action is still under investigation, these findings suggest that functional LPL is protective against the development of AD. Recent insights into the structure and function of LPL suggest that the S447X variant offers protection via increased lipoprotein uptake. Moreover, loss of function mutations in ApoC-II, which activates LPL is associated with increased AD risk. In contrast, mutations in ApoC-III, which inhibits LPL activity, is protective, further supporting the notion of a protective role for functional LPL in AD pathogenesis. Further basic studies are needed to address functional differences and to empirically determine the role of LPL in AD neuropathogenesis.

## Figures and Tables

**Figure 1 ijms-21-08338-f001:**
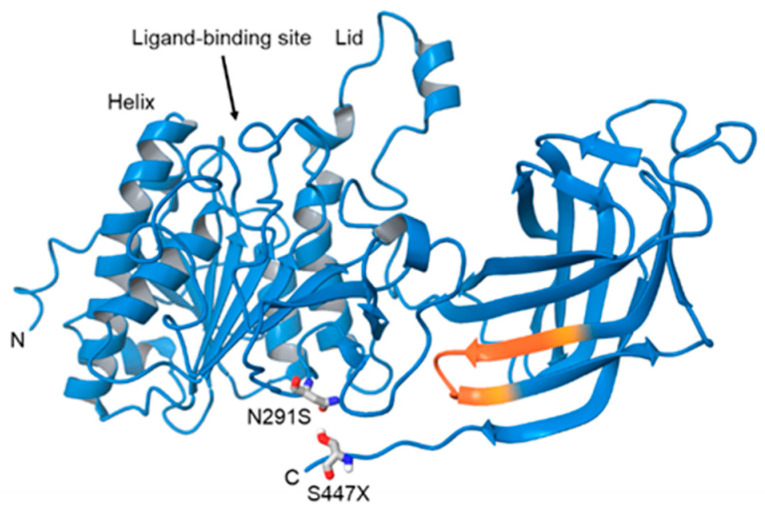
Tertiary structure of LPL. Ribbon representation of human LPL with residues of key mutation sites N291S and S447X shown in stick display style (carbons colored grey), exposed residues 405–414 upon S447X mutation are colored orange on the ribbon. The structure has been modified from the PDB: 6OB0 crystal structure elucidated by Arora et al., as residues K472, K473, S474, and G475 were added to the *C*-terminus, using the build feature of the Schrodinger Suite 2018-4. The ligand-binding (active) site situated between the lid and lid-proximal Helix. The N291S (residue 318) mutation, and S447X (residue 474) mutations are labeled. Hayne et al., proposed that truncation of the *C*-terminus via the S447X mutation, results in the exposure of a lipoprotein receptor binding site at residues 405–414 (highlighted in orange), suggesting increased lipoprotein receptor binding in this variant.

**Figure 2 ijms-21-08338-f002:**
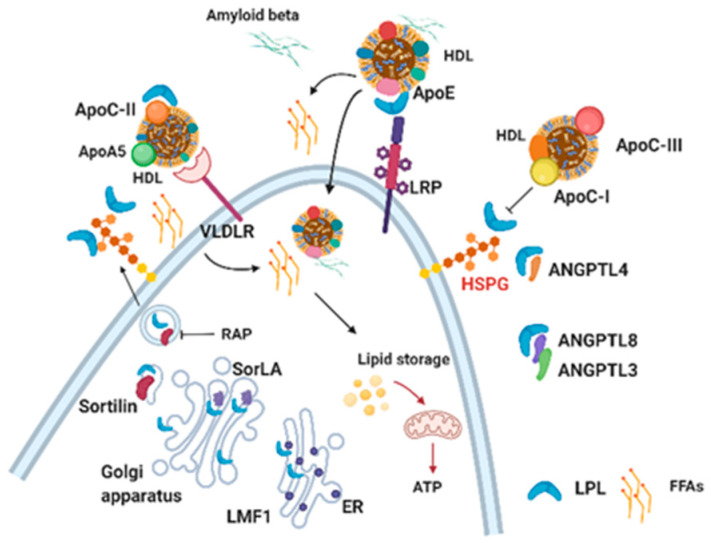
Schematic representation of LPL processing and function. LPL is chaperoned through the ER by LMF1, and sorted through the Golgi apparatus by SorLA/Sortilin. RAP can inhibit Sortilin binding, which may prevent premature secretion and affiliation with lipoprotein receptors (e.g., VLDLR and LRP). Secreted LPL is tethered to the cell surface to HSPG, where is may form a complex with lipoprotein receptors. In the brain, it is unknown whether LPL can hydrolyze the lipid components of lipoproteins, and if the canonical activators (i.e., ApoC-II and ApoA5) are involved. LPL may facilitate the endocytosis of lipoproteins via an interaction with cell surface lipoprotein receptors, providing lipids to the cell for storage, membrane formation or energy utilization. LPL also facilitates the endocytosis of Aβ, which may be HDL bound. ApoC-I and ApoC-III inhibit LPL activity. ANGPLT4 binds to the active site of LPL to inhibit hydrolytic activity. This schematic is largely based on our understanding of LPL function in microglia, and LPL processing may vary in other cells types.

**Table 1 ijms-21-08338-t001:** Functional and Clinical Characteristics of LPL Variants Associated with AD Risk

Mutation	Effect on LPL Structure or Function	AD Form	Prevalence (% of Population)	Clinical Characteristics
**N291S, Asn291Ser (rs268)**. Also known as Asn318	Mutation in exon 6 results in an A-G transmission [59]. Amino acid substitution occurs at base 318 changes the hydrophobicity and flexibility of residues 314-322 near the ligand bindings site [58].		2–5% [64]	Higher TG, lower HDL (11427211). Overrepresented in patients diagnosed with AD (5% versus 1% [63])
***HindIII* (rs320)**	T-G substitution in intron 8, at position 481 removing *HindIII* restriction site	LOAD [65]	25–39% PMID: 10830909	Increased plasma lipid profile and susceptibility to CAD. Significant association (1.75-fold increased risk) with LOAD in an Iranian population [65]. Homozygous genotypes have an increased risk of AD, more so in females [66].
***PvuII* (rs285)** Also referred to as the P+ allele.	Mutation in intron 6 within *PvuIII* restriction site. Potential cis-element of unknown significance. *PvuIII* is associated with altered LPL mRNA [67].	SAD [67]	39% PMID: 10830909	Increased risk of AD in P+ carriers. Reduced cortex cholesterol, increased NFTs and senile plaques. Combined effect of P+ and ApoE4 in the fusiform gyrus in eastern Canadian population [67].
**S447X (rs328)**. Also known as Ser447Ter.	Mutation in exon 9 leads to a C-G transversion, and loss of two final amino acids. Leads to increased receptor binding and endocytosis [68].		5–12% [63].	Decreased TG, increased HDL, reduced risk of CAD and T2D [69]. Clinically underrepresented in patients with diagnosed AD.

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
