# Peer review of "Genetic Variants of Lipoprotein Lipase and Regulatory Factors Associated with Alzheimer’s Disease Risk"

_ijms, 2020, doi:10.3390/ijms21218338_

Round 1
Reviewer 1 Report
It is always good to read about potential genetic risk candidates of Alzheimer's disease. I have a few suggestions, which may make this manuscript better prior to acceptance. A few good figures would definitely improve this review paper.
- A figure may be useful on LPL protein domains, highlighted the potential mutations, involved in AD or other neurodegenerative diseases.
- Figures on pathways of LPLs and other apolipoproteins, involved in AD and brain fucntions should be essential.
- Beside the other apoliproteins, it may also be interesting mentioning the clusterin (CLU or APOJ), which is also a potential AD risk factor.
- For chapter 4, a table would be useful, which summarizes the mutations in LPL gene, their location and their possible impact in neurodegeneration.
Author Response
Reviewer 1
“It is always good to read about potential genetic risk candidates of Alzheimer's disease. I have a few suggestions, which may make this manuscript better prior to acceptance. A few good figures would definitely improve this review paper”
We would like to thank the reviewer for carefully considering our manuscript. Your comments have given us the opportunity to improve our manuscript. The changes have been highlighted in yellow throughout the manuscript and are also detailed below.
- “A figure may be useful on LPL protein domains, highlighted the potential mutations, involved in AD or other neurodegenerative diseases”
We have now included a figure (Figure 1) which highlighted the mutations linked to AD. Importantly, we have annotated the most recent models, and the crystal structure of LPL elucidated by Arora et al., 2019. This is a major advancement over similar models that have used the pancreatic lipase homology model to annotate mutants.
- “Figures on pathways of LPLs and other apolipoproteins, involved in AD and brain functions should be essential”
A figure (Figure 2), which describes some of the major regulatory pathways for LPL has now been included. This includes LPL processing, and putative roles for LPL in lipoprotein uptake in the brain. The role of GPIHBPI has been omitted since its presence in the brain is unknown.
- “Beside the other apoliproteins, it may also be interesting mentioning the clusterin (CLU or APOJ), which is also a potential AD risk factor”
A section describing the association between CLU and AD (in brief) is now described. This includes potential interaction with LPL L140 – L152.
- For chapter 4, a table would be useful, which summarizes the mutations in LPL gene, their location and their possible impact in neurodegeneration.
A table (Table 1) summarizing the LPL mutations, their possible role on LPL function, and their clinical features has now been added.
Reviewer 2 Report
Reviewer comments
The present study by Bruce et al. discussed the genetic variants of lipoprotein lipase and several regulatory factors linked with Alzheimer's disease risk. The study point out LPL variants in Alzheimer’s Disease (AD) risk such as mutations has a deleterious effect on LPL function and expression that have been associated with increased AD risk. This review has also revealed that genetic variants in endogenous LPL activators (ApoC-II, F allele) and inhibitors (ApoC-III, G allele) can increase and decrease AD risk, respectively. Moreover, the author highlighted the significance of genetic variants of LPL and associated regulatory factors in AD risk.
The study lacks the material and method that is crucial for a paper, it is needed to add up in the abstract, and just discussing 4-5 subsection which was not found to be substantial, and the paper needed to be rewritten completely with 2-3 figures. Moreover, the authors have to modify the suggested suggestion below
- Line 20-21, change the sentence with an appropriate modification is required.
- Line 34, if the same references are using by the author then it would be better to explore the paper, not used while referencing age, people already know that it’s an age related.
- Line 55, Importantly, LPL has been repeatedly implicated in AD pathogenesis “what does it indicate?
- Line 60, grammatical mistake, please check
- Line 63, “via these processes” complicated word, please restrict these type of words
- Line 64, please provide some evidence related to that and also provide novelty of your paper
- Line 68, section 2 The section need a diagram
- Line 71-72, the information needs to be explored
- Line 119-120, any specific reason presented in the paper, i mean any other study in mice
- Line 123, a same sentence not investigated whats your role of contributing on the same
- Line 136, the author has used many times that several studies but they did not use any references there, please check
- Line 140, better to start with basic information about ANGPTL4 (suddenly comes) then its better to elaborate it
- Line 143-144, what does the line indicate
- Line 146, which disease
- Line 154-155, already discussed by you
- Line 156 how it is related to the function
- Line 183, what was the reason they presented
- Line 197-198 the author needs to write some lines before jumping to write up the important information.
- Line 208-209, Please elaborate on the paper
- Line 211, what are the methods the author have to discuss
- Line 244, Please discuss
- Line 253, please delete late-onset AD
- Line 274-275, not convincing line I found few studies
- Not convinced with the conclusion, need to be more specific
- Please check the format of references, looks like not formatted according to the IJMS
Author Response
Response to Reviewer 2
“The study lacks the material and method that is crucial for a paper, it is needed to add up in the abstract and just discussing 4-5 subsection which was not found to be substantial, and the paper needed to be rewritten completely with 2-3 figures. Moreover, the authors have to modify the suggested suggestion below”
We would like to thank the reviewer for their extremely thorough critique of our review article. We would like to respectfully point out to the reviewer that this article is a review, and therefore a “materials and methods” section is not appropriate. We have attempted to highlight the important papers regarding LPL and AD risk, in the context of a newer understanding of LPL biology, and its function in the brain. Since these have been reviewed elsewhere we have also reviewed the associations between factors that regulate LPL function and processing and AD risk; a novel aspect of the review.
We have now added three figures in line with the reviewer’s suggestions. The first figure (figure 1) highlights the LPL mutants that effect protein structure (other mutants are intronic) and highlights these on the newest LPL model rather than the pancreatic lipase homology model, which is a useful resource for those in the field.
Our manuscript has now been modified in line with the reviewers comments. These changes are highlighted throughout the revised version of the manuscript and detailed below.
“Line 20-21, change the sentence with an appropriate modification is required”
This has been modified
“Line 34, if the same references are using by the author then it would be better to explore the paper, not used while referencing age, people already know that it’s an age related”
This reference is a seminal study regarding the prevalence of AD. Since the main finding of this study relates to increasing prevalence of AD in specific cohorts, and not the molecular mechanisms leading to AD (the focus of the review), further “exploration” of this study would be beyond the scope of this review
“Line 55, Importantly, LPL has been repeatedly implicated in AD pathogenesis “what does it indicate?”
L55 – is an introduction sentence, the specific interactions between LPL and AD risk are explored in the following sentences.
“Line 60, grammatical mistake, please check”
This has been modified
“Line 63, “via these processes” complicated word, please restrict these type of words”
We have reviewed the manuscript and simplified phrases where possible
“Line 64, please provide some evidence related to that and also provide novelty of your paper”
Novelty of the paper is now described L63-68.
“Line 68, section 2 The section need a diagram”
This has been added- and has been referred to throughout the paper (see figure 1)
“Line 71-72, the information needs to be explored”
We have detailed the change in base numbering during human reference build 37. Also the discrepancy in the base numbering of the variants has been detailed in figure 1 legend, Table 1, and lines 73-78.
“Line 119-120, any specific reason presented in the paper, i mean any other study in mice”
This is the only study in mice, to the best of our knowledge
“Line 123, a same sentence not investigated whats your role of contributing on the same”
Here, we aim to highlight additional factors involved in LPL processing – so that AD associated mutations can be interpreted and disused.
“Line 136, the author has used many times that several studies but they did not use any references there, please check”
A reference has now been added.
“Line 140, better to start with basic information about ANGPTL4 (suddenly comes) then its better to elaborate it”
We agree, and actually basic information regarding Angptl4 is detailed in a previous paragraph (lines 102-111).
“Line 143-144, what does the line indicate”
Although the mechanism remains unknown, an indication has been suggested.
“Line 146, which disease”
Multiple Sclerosis has been added
“Line 154-155, already discussed by you”
This point is worth reiterating in this part of the paper, in order to facilitate the discussion.
“Line 156 how it is related to the function”
Although the mechanism remains unknown, a suggestion has been added.
“Line 183, what was the reason they presented”
Is the reviewer is asking why the variant is “over represented”? If so a suggestion, regarding the loss of function variant has been added.
“Line 197-198 the author needs to write some lines before jumping to write up the important information”
Description of all variants follow the same (and concise pattern). A description of mutation. The clinical relevance. It’s association with AD- and some mechanisms.
“Line 208-209, Please elaborate on the paper”
Although the mechanism is detailed later, more information has been given upfront (L209-212)
“Line 211, what are the methods the author have to discuss”
Additional Mechanisms of action are detailed above
“Line 244, Please discuss”
See lines 209-212
“Line 253, please delete late-onset AD”
This has been modified.
“Line 274-275, not convincing line I found few studies”
Studies, albeit inconsistent have been reported and therefore the statement that LPL has been “implicated” is appropriate in this context. Nonetheless, additional references have been added
Not convinced with the conclusion, need to be more specific
This has now been modified. The abstract has also been modified to tie together with the conclusion, as per the reviewers request.
Please check the format of references, looks like not formatted according to the IJMS
These are now in the MDPI format, as requested in the instructions for authors.
Submission Date
14 October 2020
Date of this review
21 Oct 2020 04:55:37
Round 2
Reviewer 1 Report
Authors fulfilled my suggestions, paper is acceptable now.
Reviewer 2 Report
All comments have been incorporated in the revised version of the manuscript.